# Dermatillomania: Strategies for Developing Protective Biomaterials/Cloth

**DOI:** 10.3390/pharmaceutics13030341

**Published:** 2021-03-05

**Authors:** Priusha Ravipati, Bice Conti, Enrica Chiesa, Karine Andrieux

**Affiliations:** 1Department of Drug Sciences, University of Pavia, 27100 Pavia, Italy; enrica.chiesa@unipv.it; 2Department of Pharmacie, Université Paris Descartes, 75006 Paris, France; karine.andrieux@parisdescartes.fr

**Keywords:** dermatillomania, skin picking disorder, biomaterials, polymers, physical barriers, wound healing, skin mimicking

## Abstract

Dermatillomania or skin picking disorder (SPD) is a chronic, recurrent, and treatment resistant neuropsychiatric disorder with an underestimated prevalence that has a concerning negative impact on an individual’s health and quality of life. The current treatment strategies focus on behavioral and pharmacological therapies that are not very effective. Thus, the primary objective of this review is to provide an introduction to SPD and discuss its current treatment strategies as well as to propose biomaterial-based physical barrier strategies as a supporting or alternative treatment. To this end, searches were conducted within the PubMed database and Google Scholar, and the results obtained were organized and presented as per the following categories: prevalence, etiology, consequences, diagnostic criteria, and treatment strategies. Furthermore, special attention was provided to alternative treatment strategies and biomaterial-based physical treatment strategies. A total of six products with the potential to be applied as physical barrier strategies in supporting SPD treatment were shortlisted and discussed. The results indicated that SPD is a complex, underestimated, and underemphasized neuropsychiatric disorder that needs heightened attention, especially with regard to its treatment and care. Moreover, the high synergistic potential of biomaterials and nanosystems in this area remains to be explored. Certain strategies that are already being utilized for wound healing can also be further exploited, particularly as far as the prevention of infections is concerned.

## 1. Introduction

Skin picking disorder (SPD)—also known as excoriation disorder, dermatillomania, neurotic excoriation, psychogenic excoriation, or acne excoriee—is a neuropsychiatric disorder characterized by repetitive self-excoriation/picking of skin in the absence of any underlying dermatological disease, resulting in visible tissue damage [1,2]. People with SPD have been found to engage in squeezing, scratching, rubbing, digging, and lancing into their skin, which culminate in skin lesions with varying degrees of severity. Infliction of this behavior using fingernails and fingers was implicated in most of these cases, while others have reported the use of tweezers, pins, tissues, and various other instruments [3]. The prevalent mean age of onset of SPD in individuals is reported to be approximately 13.6 years [4,5,6], with a mean duration of 12.7 years (ranging between three years and 40 years), featuring waxing and waning skin picking episodes throughout the individual’s lifetime [5]. This is suggestive of SPD possessing a chronic and recurrent nature. Individuals with SPD were observed to spend a significant amount of time (from less than 5 min/day to 12 h/day) [3,7,8,9] on picking skin at different sites on their bodies. In this regard, the face and cuticles have been reported as the most common regions picked by a majority of such individuals in several studies [10], and other regions included the chest, neck, back, rectum, or any other bodily surface that was easily accessible [3,7,11,12]. Extensive research has been conducted to understand the characteristics of skin pickers, which elucidates that in a majority of the cases (76% vs. 24%), picking episodes often occur without conscious awareness [7,13,14,15].

Skin picking behavior is either observed as a characteristic feature of various disorders, or is associated with many existing disorders, in conjunction with other symptoms. The most common comorbid conditions in individuals with SPD are major depression, anxiety disorder, and obsessive–compulsive disorder (OCD) [7,16,17,18]. Other disorders associated with skin picking behavior are autism [19,20], Prader-Willi syndrome [21,22,23,24], impulsive and borderline personality disorder [7,18], dysthymia [16,18], schizoid personality disorder [18], body dysmorphic disorder [25], mood disorder and impulse control disorder [7,26], panic disorder, social and simple phobia [16], attention deficit hyperactivity disorder (ADHD) [27], trichotillomania [28], eating disorder [7], bipolar disorder [29] as well as post-traumatic stress disorder (PTSD) [27].

The primary objective of this review was to provide an introduction to SPD and discuss its current treatment strategies as well as to propose biomaterial-based physical barrier strategies as a supporting or alternative treatment. Additionally, it elucidates the benefits of nanotechnologies in this area of application.

## 2. Methods

Table 1 enumerates the methodology used in the literature search, along with the screening strategy and the number of results (relevance). The PubMed database was searched during the f week of April 2020 with the keywords: Dermatillomania [All Fields] OR “excoriation disorder”[All Fields] OR skin-picking [All Fields] OR “neurotic excoriation”[All Fields] OR “psychogenic excoriation”[All Fields] OR “acne excoriee”[All Fields]. This yielded a total of 440 papers related to SPD, which have been summarized below. Similarly, in May 2020, the PubMed database was screened for papers published between 2015 and 2020 with the keywords “skin picking”, “skin”, “second skin”, “extra skin”, “artificial skin”, “synthetic skin”, “skin substitute”, “breathable”, “polymers”, “antibacterial”, “antimicrobial”, “cloth”, and “textiles”, with various combinations of AND as well as OR operators. Google Scholar was used to find articles with all of the words: “polymer antimicrobial biomaterial on-skin wearable aesthetic” and with at least one of the words: “water-resistant” or “waterproof” appearing anywhere within the articles.

The obtained data were processed and presented in the following order with regard to SPD: prevalence, etiology, consequences, diagnostic criteria, and treatment strategies.

## 3. Results

### 3.1. Dermatillomania as a Psychodermatologic Disorder

Psychodermatology is a domain that addresses the interaction of the mind and the skin. Dermatillomania is a psychodermatologic disorder that is both a result and a cause of psychiatric disorders such as anxiety and depression [30]. Dermatillomania is a result of a very complex interplay between the mind and the skin. Individuals experience different emotions before, during, and after skin picking episodes. Moreover, individuals have reported heightened tension and nervousness before picking, which contrasts the pleasure and relief experienced during and after picking [31]. Emotional triggers (i.e., stress, anxiety and tension); situational triggers (i.e., being in bed, reading, driving car or being alone; perceptual triggers (such as skin imperfections); tactile triggers (such as sensory intrusions such as itchiness); and environmental triggers (mirror checking) have been reported to cause picking behavior. Individuals have also reported anticipatory social anxiety as a trigger for picking [7,32,33]. Some have said that engaging in skin picking behavior helped them feel relief from anxiety, tension, discomfort, and pressure of studies [25,32,34] and reported that by doing so, they felt a sense of enjoyment and derived great pleasure [25,35]. One individual reported that it made her happier [36]. Feelings of anger, shame, guilt, self-aversion, and anxiety are also common in those suffering from SPD upon noticing the physical damage caused to themselves [37]. Some had even admitted to eating the skin they picked, which is referred to as dermatophagia [5].

### 3.2. Prevalence

The reported prevalence of SPD has been identified as 2% in dermatology clinic patients [38], constituting 71.9% of patients with psycho-cutaneous disorders visiting dermatology clinics [39], 1.19–14% in non-clinical samples [3,11,31,37,40,41,42], and 2.04–15.6% in various student populations [3,27,43,44,45,46]. SPD has been reported to occur more commonly in females (gender distribution of 87.1–94.1%) than in males, along with higher rates of skin picking and associated impact [3,11,14,38]. According to a recent retrospective study (2011–2016) conducted at a Swiss tertiary hospital in 2018, less than 5% of the patients with SPD were referred to a psychologist or a psychiatrist, while the remaining ones were administered topical or systemic anti-acne treatments by dermatologists. It may be assumed that the prevalence of skin picking disorder is still being highly underestimated, possibly attributable to the lack of awareness about SPD, in conjunction with the associated shame, embarrassment, and fear of judgement in admitting to this behavior and seeking treatment from psychiatrists [47,48].

### 3.3. Etiology

Several theories and models have been suggested in order to delineate the occurrence of SPD in individuals including the psychoanalytical, developmental, and personality theories [49] as well as the emotional regulation and frustration action models [50]. These theories and models broadly conclude that individuals engage in skin picking as a mode of coping with, or escaping from, their emotions. Moreover, independent studies have suggested that both familial [51,52] as well as genetic factors [53,54] may exert an influence on this skin picking behavior. The neurobiological basis for SPD was first explained by serotonin [49] and opioid hypotheses [55], whereby it has been associated with serotonin deficiency, increased endogenous opiates, and involvement of the dopaminergic system [49,55,56]. Decreased serotonin and increased dopamine in the ventral striatum are inherent to impulsive behavior, whereas the converse is true of less impulsive behavior [35]. Furthermore, neuroimaging studies have correlated the dysfunction of right fronto-striatal neural network [8], disconnection of white matter tract in regions involved in motor generation and suppression [57] as well as volume and cortical thickness abnormalities in both the left and right cerebral [58,59,60] and cerebellar regions [59,60,61] with impaired motor inhibitory control and maladaptive emotion regulation observed in individuals with SPD [61].

### 3.4. Consequences

Individuals with SPD pick their skin with or without conscious awareness to the extent that this behavior results in bleeding, pain, scarring, spotty faces, general disfigurement, erythematous skin lesions, persistent sores, ulcers, dermatosis, and recurrent infections that require aggressive dermatologic treatment, several courses of antibiotics to control infections, intravenous antibiotics to control sepsis, multiple hospitalizations, debridement as well as surgery and skin grafting [2,3,7,36,55,62,63,64,65,66,67,68,69]. A study reported that 61.8% of skin pickers acquired infections, of which 16% of cases required antibiotic treatments [5]. Skin picking can be a very dangerous behavior, capable of manifesting some very serious life-threatening or near-fatal conditions [66,70,71]. Other cases have demonstrated rectal bleeding and solitary rectal ulcer [12], lower gastrointestinal bleeding and anorectal disease [72], loss of penis after recurrent ulcers and multiple reconstructive surgeries [73], pyogenic myositis caused by methicillin resistant *Staphylococcus aureus* [74], picker’s or prurigo nodules [75], as well as Pilomatricoma, all resulting from skin picking [76].

Repetitive skin picking was reported to have a negative impact on quality-of-life indices of leisure, clothing choice, sexual activity, and athletic endeavors [77]. A comprehensive web-based study reported that skin picking is significantly associated with impaired physical and psychological quality of life [40,78]. According to yet another study, approximately 40% of individuals afflicted with SPD avoided social gatherings or going out to places, 54.3% refrained from getting into intimate relationships, 5% reported quitting their jobs, and 50% of them stated that skin picking interfered with performing daily activities at school, while some even ceased pursuing schooling anymore [79]. It has also been reported that individuals with SPD spend approximately $6650 in their lifetime on treatments received from mental and medical health professionals, also suggesting an immense financial impact [79]. Thus, the overall effects on physical appearance, mental health as well as the social, sexual, and occupational life that individuals with SPD often suffer from are of serious import and necessitate greater attention for the development of treatment strategies.

### 3.5. Diagnostic Criteria 

Initially, pathological skin picking was commonly reported as a feature of a variety of dermatological, medical, developmental, neurological, and psychiatric conditions [77]. However, since 2014, “Excoriation (Skin-picking) Disorder” has been included in DSM-5 under Obsessive-Compulsive and Related Disorders, establishing its identity along with a definition and diagnostic criteria [80]. The diagnostic criteria put forth for SPD in the DSM-5 are as follows: (A) Recurrent skin picking resulting in skin lesions; (B) repeated attempts to decrease or stop skin picking; (C) skin picking causes clinically significant distress or impairment in social, occupational, or other important areas of functioning; (D) skin picking is not attributable to the physiological effects of a substance (e.g., cocaine) or another medical condition (e.g., scabies) as well as (E) skin picking is not better explained by symptoms of another mental disorder (e.g., delusions or tactile hallucinations in a psychotic disorder, attempts to improve a perceived defect or flaw in appearance in body dysmorphic disorder, stereotypies in stereotypic movement disorder, or intention to harm oneself in non-suicidal self-injury) [81].

### 3.6. Treatment Strategies

#### 3.6.1. Behavioral Strategies

Habit reversal training (HRT) is a mode of behavioral therapy that comprises awareness training, competing response practice, habit control motivation, and generalization training [32,63]. HRT is known to be effective in reducing skin picking [63,82,83]; however in certain cases, it has been applied in combination with pharmacological drugs such as antidepressants [83]. It is therefore difficult to assess the effectiveness of HRT alone. In some contrasting outcomes, HRT has also been reported to be incapable of reducing skin picking [83,84]. Moreover, the association of skin picking with other comorbid psychiatric conditions should be taken into consideration, as this gives rise to further complexities in the treatment. Using HRT as a solitary treatment will, therefore, not suffice, leading to the concurrent inclusion of cognitive behavioral therapy (CBT) [32]. The latter treatment modality utilizes emotion regulation techniques as well as cognitive restructuring approaches to resolve dysfunctional thought patterns and actions that damage skin [32,85]. CBT has demonstrated great efficacy in improving skin picking behavior in several cases, as reported by Deckersbach et al. [32], and in 64.3% of 14 individuals with SPD, as reported by Schuck et al. [86]. However, it has also been elucidated that psychiatric comorbidities such as depression can interfere with CBT and cause a relapse [33], while CBT may not be as effective in individuals with developmental disorders [87]. 

Another mode of behavioral therapy is acceptance and commitment therapy (ACT), which teaches the individual to notice their thoughts and feelings underlying their engagement in skin-picking behavior. ACT uses methods to regulate these verbally-based processes, in addition to directly targeting behavioral changes [88]. It has been shown to reduce skin picking, but the positive effects were not maintained at follow up [88]. Eclectic psychotherapy with insight-oriented as well as behavioral components resulted in the healing of skin-lesions in 85% of 20 patients with dermatillomania [89]. Similarly, psychodynamically-oriented group therapies have also been designed in order to emphasize the heightened potential of effective treatment in a group of individuals facing similar behavioral problems with regard to coping with their emotions and concerned behaviors [90]. Other strategies known to reduce skin picking behavior include biofeedback procedures and hypnosis [85,91], cognitive psychophysiological approach [92], competing stimulus assessment (CSA) [93], emotion regulation strategies, cognitive reappraisal [94], cognitive hypnotherapy [95], dialectal behavioral therapy [96], self-monitored differential reinforcement of other behavior [97] as well as eye movement desensitization and reprocessing [98]. 

#### 3.6.2. Pharmacological Strategies

Currently, there are no Food and Drug administration (FDA) approved drugs specifically for the treatment of SPD. However, the common pharmacological treatment modality involves the use of opioid antagonists, antipsychotics, anti-anxiety drugs, anti-depressants, and anti-epileptic agents, as reported in various studies or trials conducted to date (see Appendix A).

Antidepressants such as selective serotonin reuptake inhibitors (SSRIs) that increase the serotonin levels in the brain by inhibiting their reuptake are considered the first line of treatment for skin picking involving compulsive features or comorbid anxiety and depressive disorders [99]. Antipsychotics have demonstrated the ability to reduce skin picking in individuals through their activities as both dopamine and serotonin receptor antagonists. Moreover, administering a combination of antipsychotics and antidepressants has also been shown to result in a reduction in skin picking behavior, suggesting that balancing both the serotonin and the dopamine levels at different regions of the brain is necessary for controlling the skin picking behavior (Appendix A). Opioid antagonists such as naltrexone, which binds to opioid receptors in the CNS and blocks the effects of endogenous opiates released during skin picking behavior, are one such category of drugs [55]. Similarly, anti-epileptic drugs that increase gamma-aminobutyric acid (GABA) activity and suppress the release of excitatory glutamate are also known to be effective in certain cases. Reports have suggested that glutamate can affect activity in the key reward circuitry, which may decrease the craving to indulge in reward-seeking compulsive skin picking behavior [84,100,101]. Consequently, anti-epileptic drugs can exert a positive influence on controlling skin picking behavior. 

#### 3.6.3. Alternative Pharmacological Strategies

Lithium carbonate is a mood stabilizer that acts by modulating glutamate receptors and has been shown to improve skin picking behavior as well as acne excoriee in an individual [102]. Other drugs that function on a similar basis include *N*-acetyl cysteine (NAC) [84,101] and Riluzole [100]. NAC is the only drug that has demonstrated the most promising results in treating SPD thus far. To this end, an open label pilot study established its efficacy in PWS individuals, wherein 71% of them exhibited complete resolution of skin picking [103]. Additionally, in a 12-week randomized double-blind trial, administration of NAC resulted in improved profiles for 47% of the individuals, compared to 19% who were given a placebo [104]. Several other cases where NAC has been linked to the reduction or prevention of skin picking behavior have also been reported [84,101,105,106,107]. Despite these promising results, NAC may often be poorly tolerated by individuals due to its mucolytic properties [108]. In certain cases, a combination of both behavioral and pharmacological therapies is applied for better results, as presented in Appendix A.

The same class of drugs that have been reported to be successful in the studies above-mentioned have contrastingly proven unsuccessful in some other cases, or even induced or aggravated skin picking, as reported in Appendix A. However, despite the positive effects observed in certain individuals, it has also been observed that discontinuation of the drug caused recurrence of the behavior (Appendix A). More comprehensive studies, along with sufficient follow up and tolerance data, are thus required to efficiently assess the efficacy of these drugs in treating SPD. 

#### 3.6.4. Other Strategies

Internet-based interventions such as SaveMySkin have been developed to offer far-reaching and easily accessible support in the form of information, exercises, and chat counselling sessions [109] that showed substantial reductions in skin picking severity in the intervention group compared to the control [110]. Another Internet-based treatment via StopPicking.com for self-injurious skin picking has also demonstrated a significant reduction in skin picking frequency and symptom severity [111]. Technology-based interventions like smart watches such as the Keen bracelet [112] and the Tingle wearable [113] have also been developed, but data on their efficacy is insufficient. Moreover, the use of physical strategies such as contingent gloves, as reported in sensory impaired adolescent [114], bubble helmet in a disturbed autistic child [115], and use of gloves and masks [19] have been implicated in the reduced frequency and intensity of skin picking, self-biting, and self-injurious behavior, respectively. However, very limited research has been conducted on the effectiveness of physical strategies in SPD.

A meta-analysis on psychiatric treatments for SPD suggests that, although the behavioral and pharmacological treatments utilized in SPD to date have exhibited definite benefits, there is a lack of strong evidence to support any specific treatment or suggest its unique benefit [116]. The existing treatments are only partially successful as they lack the feature of “one size fits all”. Not all individuals respond to a single treatment, thus making the treatment of this disorder particularly challenging. This not only suggests the necessity for understanding the etiological and neurobiological basis of SPD better, but also for the development of alternative, new, and effective strategies for its prevention and treatment. 

### 3.7. Presenting the Need for an Alternative Treatment Strategy

Since dermatological treatment has been shown to be generally ineffective [65,76,117,118], while behavioral and cognitive techniques require high levels of motivation and commitment from individual’s part for success [49] and pharmacological therapies do not seem to be universally effective, new strategies or products are required. In a study conducted with 760 individuals suffering from SPD, among the possible medications and treatments received, only 11.8% and 4.3% of them reported their skin picking behavior to be moderately or significantly, respectively [10]. In another study, 81% and 84% of the individuals with skin picking reported using clothing and cosmetics, respectively, to camouflage the damage inflicted on themselves [77]. Individuals reported the use of makeup to cover scabs and blemishes caused by skin picking [33]. Similarly, 85% of the SPD cases in a different study attempted to conceal the effects of skin picking with the use of makeup/hairstyles/clothing [79]. Individuals have also tried applying gloves, band-aids, tapes, and aversive tasting substances on the picking sites to prevent this behavior [32,88]. Individuals with this disorder have been reported as most likely to raise aesthetic concerns and spend a median of $400 (range = 50–$2000) on dermatologist fees, along with a median of $40 (range = 0–$500) within the past three months or $160 per year on products to conceal the effects of skin picking [47,79]. However, existing research on physical barrier strategies to eliminate behaviors like skin picking is limited.

In an online survey conducted by the authors of this paper (see Appendix A), 52 out of 112 responses were considered for further analysis. Of these 52 individuals, 86.5% (*n* = 45) reported having dermatophagia or dermatillomania, or both. With the treatment strategies currently available, only 11.1% reported being successful in controlling their behavior, while the remaining reported being only partially successful (48.9%) or unsuccessful (42.2%). Approximately 86.7% of the 45 individuals thought that there should be a better treatment modality than those that already exist. This clearly represents a patient need and market demand for new products to support the treatment of dermatillomania and its varying side-effects.

### 3.8. Proposing Biomaterial-Based Physical Treatment Strategies

Following the results gathered from the bibliographic research and the online survey, we investigated potential SPD biomaterial-based physical treatment strategies. 

Some of the characteristics that an ideal physical barrier product for SPD should possess are mentioned in Table 2. These properties are primarily based to overcome consequences relating to the characteristics of the disorder, its impact on skin, and the emotional turmoil individuals with SPD go through. The underlying idea is to create a product like an artificial or synthetic skin that can function as a protective layer on skin and possess suitable mechanical properties to resist the force of tear similar to the force exerted by individuals when engaging in skin picking behavior. It is also very important for the material to have good adhesive properties to skin, while simultaneously being capable of facilitating its removal without damaging the skin.

The skin is the largest organ of the human body with a surface area of 1.5 to 2 square meters and plays a crucial structural as well as functional role in keeping the human body healthy [119]. It functions as a barrier between the human body and the external environment, whereby it protects the body from external physical and chemical factors and functions as a first line of defense against pathogenic microorganisms. It is a complex structure comprising the outermost layer, epidermis, the intermediate dermis, and the innermost hypodermis, which are all involved in maintaining body homeostasis [120]. The barrier function is primarily attributed to the epidermis, which is composed of stratified epithelium consisting of flat keratinized cells at the surface (stratum corneum) [119]. In individuals with skin picking behavior, several of these functions and properties of skin can be compromised. Skin is structurally robust; the regulation of its mechanical properties essential to the protection of vital organs from physical trauma. The mechanical properties of skin vary at different regions of the body, according to Langer’s lines of orientation [121]. In addition, skin thickness also varies across the body, from 0.5 mm in the eyelids to 2.0 mm on the back. Skin thickness not only varies from one person to another, but also exhibits intrapersonal divergences. Therefore, in order to develop skin mimicking materials, it is important to incorporate tunable properties capable of adapting to such variations as well as to different regions of the body [122]. Table 3 lists some of the properties of skin that must be considered while designing skin-mimicking materials or products. 

Physical damage to the skin caused by skin picking behavior can compromise the integrity of this organ. It has been reported that the mechanical integrity of stratum corneum in vivo has a breaking threshold of 0.204–0.408 kg/cm^2^ in healthy individuals [129]. Stripping experiments indicated that the barrier function of this uppermost layer of epidermis was altered under the specific experimental conditions [129]. Mechanical properties such as extensibility, elasticity, and hysteresis were altered upon stripping, thus compromising skin barrier function. Additionally, stripping resulted in a six-fold increase in the moisture vapor transmission rate (MVTR) [129]. An intact barrier function is reflected by low MVTR values, while any disruption in the barrier integrity causes an increase in MVTR due to heightened permeability. Although there is a lack of thorough investigation on the changes in mechanical integrity and barrier functions that occur during skin picking, it may be assumed from the stripping experiments that individuals suffering from SPD have a compromised barrier function since they were found to pick skin pervasively through the epidermis, and in some cases, reached the dermis layer, presenting much more damage [66]. Timely medical intervention for the wounds inflicted by skin picking behavior is critical to prevent infections or scarring. Queen et al. [130] reported an MVTR of 1000 to 2500 g/d to be an adequate level of moisture to prevent dehydration or exudate accumulation. Higher MVTR values can make the wound dry and lead to scarring, whereas a lower MVTR would result in the accumulation of exudates and increase the risk of bacterial infection [131]. Therefore, breathability of materials with appropriate MVTR is necessary to not only protect against infections, but also to prevent skin maceration, since the sight of irregular skin resulting from the same can be another trigger for skin picking.

Due to the lack of investigation about specific details on SPD such as with regard to the force exerted by the fingers on skin during skin picking episodes, we introduced some of the biomaterials or products identified in the existing literature that have been summarized in Table 4. The products selected may be further modified and improved upon in order to develop optimal treatment strategies for skin picking, while maintaining mechanical properties closely resembling those of skin and allowing them to be tunable for future applications. A total of six different products in research were shortlisted as potential candidates for supporting treatment of SPD. Their compositions and production techniques were synoptically compared and presented along with suggested future studies or improvements for use in SPD (Table 4).

In 2016, Yu et al. [125] reported the synthesis and application of an invisible wearable, skin-conformable crosslinked polysiloxane polymer that could be topically applied. Polysiloxane is an elastomer with viscoelastic properties that is commonly used as a skin-mimicking material [123]. A siloxane polymer was chosen here due to its established safety profile and tunable mechanical properties such as flexibility, elasticity, elongation, toughness, moisture/oxygen permeability, and adhesion to the skin via van der Waals interaction. The so- called ‘XPL technology’ that was developed is a two-step topical delivery system in the form of a cream, which can be safely and easily deposited on skin in situ. The formulation is very well designed, with each of the components either being considered as generally regarded as safe substances (GRAS) or having the necessary safety profile for leave-on skin application. The in situ formation of the polymer on the skin is based on a platinum-catalyzed hydrosilylation chemistry. This remarkable technology has demonstrated the capability of restoring normal skin aesthetics with excellent elastic recoil, flexibility, and elongation and can be worn for 16 hours. Additionally, the water-resistant and detergent-resistant properties of this technology make it even more suitable for regular wear, causing minimal interference with everyday activities. It was reported by Li et al. [127] in 2020 on one component of waterborne in vivo cross-linkable polysiloxane coatings for artificial skin, which are an improvement on the XPL technology with regard to its formulation strategy. Authors have also reported the development of a single cream containing all components, as opposed to the prevalent two-step topical delivery system, where one cream contains the polymers and the other contains the platinum catalyst. In this new formulation strategy, a single cream containing the polymer components and the platinum catalyst was made by introducing the catalyst in the form of capsules made of ethyl cellulose. However, with this new formulation strategy, polymer curing on the skin took about 30 min, which was longer compared to that with the XPL technology (~2 min). In terms of daily usage, it would be quite bothersome for an individual to apply the cream and wait for 30 minutes for it to form a polymer layer on the skin. Although it has been previously mentioned in the paper of Li et al. [127] that the application of the two creams like that of the XPL technology needs to be highly skillful, a separate study on the use of XPL technology in atopic dermatitis patients showed that all patients found it easy to apply as well as remove. This study also suggested that XPL technology could be worn while bathing/showering, allowing it to be the most suitable candidate for everyday use, while also reducing the risk of skin maceration [132]. Silicones are widely used as skin protectants due to their biocompatibility, and XPL, which is a silicone-based technology, has all the pre-requisite properties to be used as a supportive treatment strategy in SPD. This strategy may prevent individuals from directly picking their skin, which is essentially substituted with the mechanically stable second skin. Moreover, the introduction of antimicrobial properties into the XPL technology may make it an excellent adjuvant treatment strategy. 

Another strategy that may be used to prevent infections of the already wounded regions due to skin picking is the use of nanoparticles with antimicrobial properties such as silver nanoparticles, which are known to possess wide-spectrum antimicrobial properties. McLaughlin et al. [133] developed sprayable peptide-modified silver nanoparticles that can function as an anti-infective and anti-biofilm barrier. In order to obtain stable and non-toxic nanoparticles, silver nanoparticles were capped with a thiol-modified LL37 antimicrobial peptide and were chemically crosslinked to collagen. These surface-grafted silver nanoparticles with LL37 and collagen were used to prepare a colloidal suspension, which when sprayed formed a AgNP@LL37 collagen film. This study demonstrated the effectiveness of the spray on the Gram-negative *Pseudomonas aeruginosa*, and the same anti-microbial peptide, LL37-SH was also found to be effective against Gram-positive *Staphylococcus aureus* [134]. This sprayable formulation exhibited minimal organ infiltration upon application to full thickness wounds in mice, thereby making it ideal for safe topical application [133]. This formulation may be used to prevent infections by spraying onto raw skin exposed due to skin picking behavior. However, a safety evaluation of this formulation needs to be conducted in humans.

The use of self-healing materials, which are considered smart materials and mostly include polymers or elastomers, is yet another potential strategy for treating SPD. These materials are capable of repairing themselves upon damage, thus allowing this property to be usefully exploited for the development of patches or healable coatings, or other apparels that can be worn on skin to prevent skin picking. They should also facilitate self-healing or reuse of the material, if marred due to skin picking behavior. Some of the approaches to achieve self-healable properties include the incorporation of encapsulated-monomer systems, reversible covalent bond formation, or supramolecular self-assembly (facilitated by non-covalent bond association) [134]. Feula et al. [139] reported the development of an adhesive supramolecular polyurethane elastomer that was self-healable at room temperature. Similarly, Lei and Wu [135] reported the development of self-healable acrylamide-based hydrogel with the aim of developing biomimetic skin-like ionotronics for wearable smart applications. This hydrogel has demonstrated a remarkable and wide spectrum of mechanical properties that can mimic natural skin. Its flexible reconfiguration ability, self-healing ability within two hours as well as recyclability make it a suitable material for designing self-healing adjuvant apparels that may be useful to individuals with SPD. However, in vivo tests on skin still need to be performed to further assess the skin biocompatibility of this material. Wang et al. [136] reported a highly self-healable plasma amine oxidase induced dual network epsilon poly L-lysine (EPL) hydrogel with robust mechanical properties and broad-spectrum antimicrobial properties, marking it as another desirable material for application in supportive treatment strategies for SPD. EPL is a natural antimicrobial cationic peptide and GRAS with antibacterial and anti-fungal properties [136]. In comparison to the hydrogel reported by Lei and Wu [135], this hydrogel-based material with adhesive properties and proven biocompatibility is superior in terms of anti-bacterial and wound healing properties, making it better suited to the development of patches or tapes. 

The synthesis or production techniques used for the above-mentioned research products are well known and industrially established, comprising methods such as homogenization and mixing for emulsions and suspensions, random and radical polymerizations of polymers, or the preparation of hydrogels, which make future translation more feasible (Figure 1).

Similarly, other appropriate emerging techniques for industrial production include electrospinning and 3D printing. Zhao et al. [137] reported the synthesis of eco-friendly, water proof, and breathable polyurethane membranes with antimicrobial properties, obtained by incorporating silver nitrate salt into the polyurethane (PU) and PU (with short perfluoro butyl chain, C4FPU) solution used for electrospinning. The excellent tensile strength of these membranes fabricated via the electrospinning process makes them suitable for the development of protective garments (Figure 2C). A novel polyurethane elastomer (C4FPU) with a double terminal short perfluoro butyl (−C_4_F_9_) chain has been used in this study due to its low toxicity and low bioaccumulative potential, which culminates in the production of eco-friendly membranes possessing hydrophobic properties. Further research on the wash and reuse of these membranes as well as the corresponding shift in mechanical and antimicrobial properties needs to be conducted in order to develop these into competent protective clothing for individuals with SPD. Like electrospinning, 3D printing has been gaining significant attention lately for large-scale production in medical fields. Muwaffak et al. [138] have reported a strategy of 3D scanning and 3D printing of patient-specific flexible wound dressing consisting of a FDA approved polymer, polycaprolactone (PCL) (Figure 2D). Polycaprolactone is a biodegradable and biocompatible polymer that has been widely used for various applications such as drug delivery, tissue engineering, and preparation of scaffolds as well as in-wound dressings. PCL filaments loaded with silver or copper salt were utilized in the 3D printing of anatomically fitting wound dressings with antimicrobial properties. These wound dressings exhibited slow and prolonged release of silver or copper ions, which may be considered advantageous with regard to limiting the frequency of changing the dressing. This controlled release is attributed to the entrapment of silver and copper ions in the PCL matrix, which acts as a barrier to the release of ions due to the slow penetration of water into the matrix. Dressings for anatomically complex areas via personal 3D scanning and printing may provide more comfort and aesthetic value. However, these assumptions need to coincide with the patient’s responses in future investigations. 3D printing is a very new technique and requires a lot of optimization of settings for the production of 3D products. Several parameters such as filament thickness or diameter, layer height, speed of extrusion and travelling, number of layers or shells, printing time as well as temperature control influence the end product and its properties. This emerging technique can be utilized even for personalized or largescale printing of products. Nonetheless, initial optimizations and production parameters must be thoroughly investigated to facilitate the high quality and consistent production of desired products across batches. Sterility of the physical barrier products lacking antibacterial properties and intended for application on wounded skin is yet another important factor to be taken into consideration.

## 4. Discussion

SPD is a complex disorder that involves both behavioral as well as physical impairments. The identification of other comorbid neurological conditions in individuals with SPD is of utmost importance as it can interfere with its treatment. For instance, individuals with Attention deficit hyperactivity disorder (ADHD) who were administered methylphenidate developed skin picking behavior [43]. It is therefore possible to unintentionally worsen the symptoms of individuals with SPD when given methylphenidate for ADHD, thus emphasizing the need for taking comorbid conditions into consideration when designing therapeutic interventions. Conversely, clinicians must also consider the presence of dermatillomania when treating other neurological disorders. 

Skin picking can be a very dangerous behavior. It is very important to consider the possibility of skin picking disorder when examining skin lesions. For example, after several failed attempts at identifying the source of bacteremia in a 54-year-old African-American with a history of diabetes mellitus, it was realized that the patient’s compulsive skin picking behavior led to the infection. The skin lesions and ulcers on the patient’s neck, fingers, and legs might have misled the physicians to associate them with diabetes or any other disorder, but identifying the root cause of the skin lesions and referring the patient to psychotherapy ultimately contributed to the significant improvement of the patient’s condition. Disregarding skin picking disorder as a cause of infection in this case would have possibly resulted in serious complications in the patient due to his irresistible skin picking behavior [44]. Therefore, raising awareness of SPD among physicians and patients is necessary to ensure better diagnosis and treatment of SPD as well as symptoms and disorders secondary to SPD.

This report aimed to propose less restrictive and more protective biomaterial-based equipment to prevent self-inflicted injury caused by skin picking. We surmised that physical barrier strategies that could reduce or mask the injuries due to skin picking and also protect individuals from damaging their skin when engaging in this behavior would prove more ideal in all situations, given the fact that individuals indulge in this behavior both consciously and subconsciously [52]. 

The opinions of individuals with SPD must also be given due consideration when designing treatment strategies. In a survey we conducted, approximately 42.2% cases reported that they derived pleasure from engaging in the behavior, but disliked the harm it did to their bodies. Another 42.2% reported that they did not seem to enjoy engaging in skin picking behavior (See Appendix A). We believe that using physical barrier strategies as a method of response prevention and response substitution in treating the chronic, recurrent, and treatment-resistant SPD may prove effective and could be a suitable support to either the other forms of therapy or the “one size fits all” strategy. 

Natural polymers, semisynthetic polymers, synthetic polymers, or various combinations of these have been extensively used in generating scaffold materials for skin substitutes [140], for wound healing and dermal reconstruction [141], in wounds and burns dressing [142], and as skin models to simulate the physical properties of skin [123]. Table 5 describes some of the polymers widely used to mimic or simulate various skin properties.

In this review, we presented products under research that are prepared from polymers such as silicones (polysiloxane), polycaprolactone, polyurethanes, collagen, and combinations of synthetic polymers. The formulation strategies involve the use of biocompatible polymers to simulate the physical, mechanical, and surface properties as well as physiological conditions of the skin; use of microspheres to simulate the optical properties and refractive index of the skin; and use of silver nanoparticles, salts, and peptides (epsilon poly l-lysine and LL37) to mimic the anti-bacterial properties of skin, as also reported in the examples of Figure 2.

In cream 1 of their XPL technology, Yu et al. [125] used fumed silica in their reactive polymer blend (RPB) to confer mechanical toughness to the polysiloxane film. They elucidated that fumed silica increased leather adhesion (78 N/mm), fracture strain (over 800%), and tensile modulus (0.48 MPa), in addition to increasing the viscosity of RPB to an extent that topical spreadability of RPB on skin was poor (600 Pa/s, measured at 0.5 s^−1^). This problem was solved by developing a new water-in-silicone emulsion system with RPB and plasticizers in the external silicone phase, in conjunction with an aqueous internal phase thickener to allow appropriate shear thinning effects for uniform and easy deposition of the siloxane phase on skin. However, these shear thinning effects are observed at higher shear rates above 5 s^-1^ (which declines with the shear rates reported for topical cream application), suggesting that the higher viscosity of the cream at low shear rates reduces gravity-driven sedimentation and supports longer shelf-life. Cream 2 consisted of the platinum that catalyzed the crosslinking of RPB to form polysiloxane film. Nylon microspheres surface-treated with isopropyl titanium triisostearate (diameter 8 μm, refractive index 1.54) were used as light scattering particles in cream 2 to mimic the optical properties of the skin. These surface-treated microspheres are known to be used in cosmetics due to their superior skin affinity [143]. Use of light scattering particles to simulate absorption and scattering properties of tissues is a common strategy, some other reported particles that have been used for this purpose include monodispersed polystyrene and titanium dioxide particles [123]. A one component strategy containing the Karstedt catalyst capsules, along with a vinyl dimethicone emulsion and hydrogen dimethicone emulsion in an aqueous continuous phase has been developed by Li et al. [127]. The key modification in this formulation is the use Karstedt catalyst capsules with an ethyl cellulose shell synthesized via the solvent evaporation method. Obtaining smaller capsules in order to ensure uniform distribution of the catalyst in the formulation during topical application was the primary goal. Core-to-shell mass ratio (C/S), concentration of the catalyst, and PVA concentration (used as an emulsifier to reduce interface energy between oil and water phase) were the critical factors involved in regulating the size of the catalyst capsules. Higher C/S mass ratio resulted in holes on the surface of the shell due to the thinness of the shell, which led to solvent evaporation causing the formation of holes. In contrast, capsules with a lower C/S mass ratio possessed thicker shells that provided mechanical strength to the capsules, but deteriorated the release of the catalyst. A C/S mass ratio of 1:1 was chosen as the ideal in conjunction with 4 wt.% catalyst concentration and a PVA concentration of 25 wt.%, in order to obtain capsules having a smaller Z-average size of 250 nm. Polysiloxane film forming components and the Karstedt catalyst capsules within a single formulation were well isolated from each other by the water phase, which provides sufficient stability to the preparation. However, the Karstedt capsule integrity is what ultimately determines the long-term storage stability. The film is formed (curing time 10–30 min) upon application by the evaporation of the water phase and merging of the vinyl dimethicone with the hydrogen dimethicone droplets to form crosslinked polysiloxane film, catalyzed by the release of the Karstedt catalyst from the capsule into the oil droplets [127]. Both these studies involved in vivo testing in human subjects and ruled out safety concerns of the formed film in contact with skin [125,127]. However, long-term stability testing in different climatic zones is necessary for further translation of the capsule-based formulation strategy as leaching of the platinum catalyst from the capsule is a valid and possible concern during storage. 

Antibacterial barrier properties were achieved by using modified silver nanoparticles, silver nitrate salt, copper sulfate salt, and natural polypeptide epsilon poly l-lysine (EPL). Due to emerging multidrug-resistant bacteria, silver nanoparticles (AgNPs) with broad spectrum antibacterial properties have garnered much attention. The bactericidal properties arise from the interaction of the silver ions with the thiol or amino group of proteins, nucleic acids, and cell membranes as well as by causing the generation of reactive oxygen species (ROS) [144]. However, silver resistance genes in bacteria against ionic silver have also been identified to exist [145]. Whether the AgNPs act as a depot of silver ions and release them, or the nanoparticles as a whole exert a bactericidal effect is still unclear, necessitating further modifications of these AgNPs in order to achieve better stability and antibacterial effects. To this end, using protein capped AgNPs has been shown to confer colloidal stability to the silver nanoparticles [146]. Surface-grafting AgNPs with antimicrobial peptide LL37, as reported by McLaughlin et al. 2016 [133], provided stability as well as prevented surface oxidation of AgNPs. In this case, the bactericidal properties were not only due to the AgNPs, but also the antimicrobial peptide interacting with bacteria. These LL37 peptide capped silver nanoparticles were prepared by exchanging citrate with LL37-SH from citrate capped AgNPs. This exchange resulted in an approximately 200-fold increase in hydrodynamic size (from 4 to 750 nm). Subsequently, these peptide capped AgNPs crosslinked with collagen were developed into a spray, which when sprayed formed a collagen film embedded with LL37 modified AgNPs. The formulation exhibited no toxicity in both in vitro as well as in in vivo experiments. The minimal organ infiltration achieved by this formulation, when compared to AgNPs crosslinked to collagen, may be attributed to the increased size of the LL37 peptide capped AgNPs coupled with the restricted release from formed collagen layer. This is quite an impressive accomplishment, especially considering the development of safe topical products, wherein non-targeted organ infiltration is undesirable. However, further testing of hemocompatibility and toxicity in humans is necessary for establishing a detailed safety profile before developing it into a product. Antibacterial properties exhibited by products reported by Zhao et al. [137] and Muwaffak et al. [138] are attributable to the release of silver ions from a novel PU elastomer-based nanofiber matrix and polycaprolactone matrix, respectively, containing silver nitrate salts. PCL wound dressings with embedded silver nitrate salts have demonstrated slow and prolonged release of silver ions from the matrix, which is explained by the time taken for the silver ions to move to the surface of PCL from the matrix prior to their release. PCL matrix acts as a barrier for the release of ions, thereby causing a slow release of silver ions. Similarly, copper ions have exhibited antibacterial activity via the inhibition of biological activity by altering proteins, causing membrane lipid peroxidation, and causing plasma membrane permeabilization. Copper ions were also observed to possess healing properties by promoting angiogenesis, making it a satisfactory choice for use in wound dressings [138]. However, as reported by Muwaffak et al. [138], copper salt embedded in the PCL matrix exerts antibacterial effects, but at higher concentrations compared to the silver salt-loaded PCL. This could potentially be due to the even slower release of copper ions compared to silver ions. The slower release of copper ions from the PCL matrix could be attributed to the lower solubility properties of copper in water when compared to silver [138]. Consequently, this indicates that silver is a better bactericidal agent compared to copper. To allow translation of the results obtained by Zhao et al. [137] and Muwaffak et al [138], in vivo toxicity assessments of the products need to be performed further.

Breathability and waterproof properties depend on the structure and nature of components comprising the product. These properties can be achieved by using hydrophobic microporous materials with pore sizes smaller than the finest water droplet, but larger than a water vapor molecule. Zhao et al. [137] synthesized polyurethane nanofibers using PU to confer mechanical properties, and incorporated polyurethane with double terminal perfluorobutyl (–C_4_F_9_) (C4FPU) chains acting as hydrophobic segment to confer waterproof properties. Electrospun fibers assembled into interconnected networks provided pore spaces, making it suitable for the development of breathable and waterproof products. Additionally, the introduction of silver nitrate salt into the electrospinning mix increased its conductivity, resulting in the formation of thin nanofibers due to elevated whipping and spinning fluid during the electrospinning process. The diameter of these fibers decreased from 718 nm (without AgNO_3_ and at 2% C4FPU) to between 460 nm and 350 nm with increasing silver nitrate concentration. Addition of silver nitrate decreased the d_max_ (1.35 to 1 μm) as well as the porosity (40.5% to 30.2%) of the electrospun fiber membranes. The excellent water-resistance and breathability properties can be controlled by regulating the electrospinning process parameters, type of components used, and their concentrations, in order to adjust the structure and porosity for the desired applications. Moreover, in vivo testing must be performed in order to assess the antibacterial efficacy and membrane compatibility with skin, and to allow future translations into applicable products. 

The nature and concentration of polymers (hydrophilic or hydrophobic), production method as well as size and structure of material or nanoparticles, all influence the final physical, mechanical, optical, antibacterial, physiological, and adhesive properties of the end product. These, in turn, determine the suitability, effectiveness, and applicability of such physical barrier products for their use in treating SPD.

Extensive evaluation of the proposed biomaterials must be conducted with regard to properties such as their duration of wearability, reusability, removability, comfort, skin-compatibility, and safety; alterations in the natural skin microbiome and its impact on health in the case of antibacterial wearables as well as their social acceptability and effectiveness in preventing skin picking and infections. Skin thickness, mechanical properties, and skin appendages like hair vary from region to region on the body and need to be taken into consideration when applying topical products like XPL. It is also mandatory to ensure that the sensory functions of skin are maintained and not compromised with the use of physical barrier strategies. While all these parameters are concentrated on treating SPD, it is also important to consider that individuals with SPD may suffer from comorbid dermatophagia, whereby individuals chew or gnaw at their skin. This may give rise to further complications when individuals with dermatophagia subconsciously gnaw at their skin covered with physical barrier protective equipment. This review primarily focused on dermatillomania, and not dermatophagia. However, Houston-Hicks et al. [147] previously reported on the usage of PLAY HANDS protective gloves for children with developmental disorders such as cerebral palsy, who injure themselves by biting their hands due to sensory and pain issues. They developed protective 3D-printed hand wear intervention using biodegradable/bioabsorbable polymers such as polylactic acid or high to ultra-high molecular weight polyethylene (UHMWPE), which are non-toxic and are most commonly used in food packaging materials. These materials possess certain necessary mechanical properties that ensure the products are less likely to cause damage to the teeth while chewing. To the best of our knowledge, this is the first ever study to develop a physical barrier strategy specifically for dermatophagia, with the concept of designing cosmetically appealing hand wear for children. While physical interventions specifically for dermatophagia have been developed previously, no such interventions for dermatillomania are under research. This is also the first ever review to propose biomaterial or nanotechnology-based physical barrier strategies as an alternative treatment strategy or a supportive strategy for the treatment of SPD. We highly recommend the development and evaluation of physical barrier strategies possessing all the ideal properties, as described above, for the betterment of the lives of those individuals suffering from skin picking disorder and in urgent need of improved treatment strategies. Development of products with the mentioned ideal characteristics for SPD may not only be used for SPD but in other disorders such as Prader–Willi syndrome or other neurodevelopmental disorders where skin picking manifests as a maladaptive behavior. Moreover, it may also be used for preventing contact dermatitis, cosmetic applications, topical drug delivery, other biomedical applications, and for smart wearable applications, among innumerable other potential uses.

## 5. Conclusions

SPD is a mental health disorder that needs urgent attention, treatment, and care. The added value of biomaterials and nanosystems in this domain remains to be explored. In some ways, the treatment of SPD resembles the strategies that are already in place for wound healing, particularly as far as the prevention of infections is concerned. The large number of nanotechnology-based wound healing medications described in the existing literature suggests that these technologies could improve the loading, release, and stability of different kinds of antimicrobials. Some examples of how nanotechnology could help in supporting SPD treatment have been described in this review. On the other hand, specific nanotechnology-based nanocomposites could help in improving the mechanical as well as tear-resistance of medications, making them difficult to be removed or scratched. Development of cost-effective and reusable products, along with the design of mobile applications to scan skin picking regions and obtain personalized aesthetically appealing or skin tone matching medications or apparel capable of restoring barrier integrity, will also significantly improve the social comfort and everyday lives of individuals suffering from SPD. The production of self-applicable remedies based on simple polymer formulations could prove beneficial to patients with regard to improving their quality of life, while also motivating them toward greater independence in the management of SPD, thus promoting their empowerment. The skin protective products based on biomaterials are either medical devices or cosmetics. Increasing the research in this area will unlock new markets for medical devices and cosmetic industries. Moreover, the efficacy of these products will reduce the use of prescribed drugs, along with their associated side effects and costs.

## Figures and Tables

**Figure 1 pharmaceutics-13-00341-f001:**
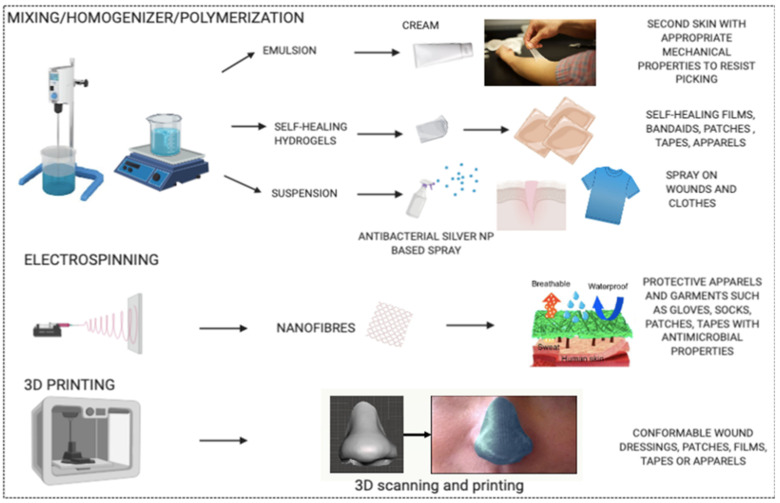
Graphical illustration of the proposed physical barrier strategies and their manufacturing techniques. Modified and assembled via the Biorender software and the Servier medical art website.

**Figure 2 pharmaceutics-13-00341-f002:**
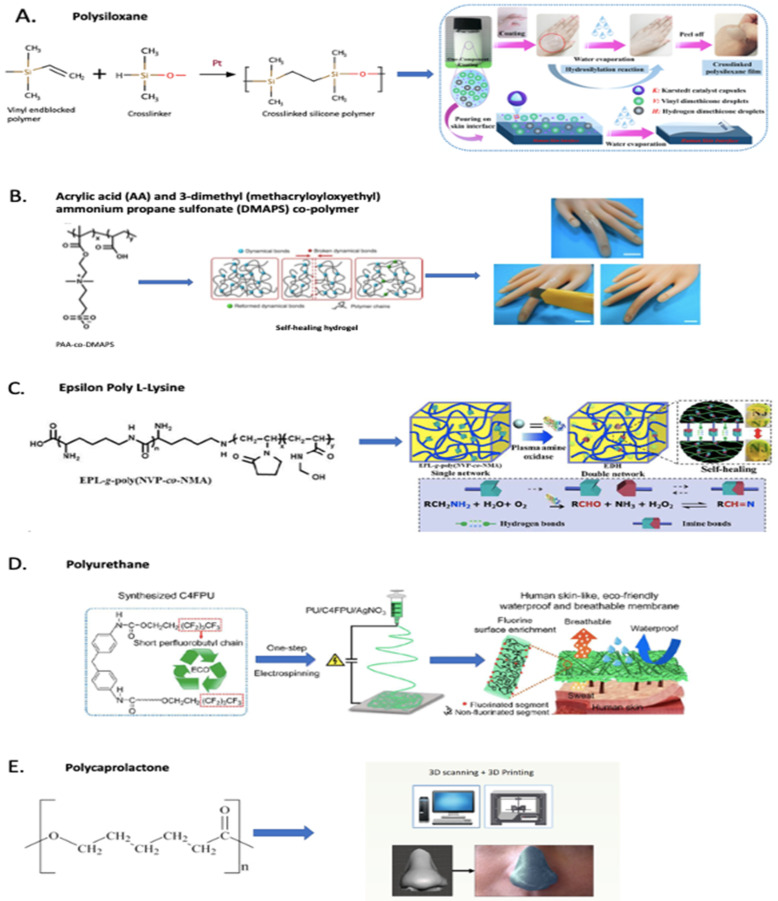
Examples of synthetic polymer formulation strategies: (**A**) Polysiloxane-based formulation for second skin [125,127]; (**B**) Acrylic acid (AA) and 3-dimethyl (methacryloyloxyethyl) ammonium propane sulfonate (DMAPS) co-polymer-based self-healing hydrogel [135]; (**C**) Epsilon Poly l-Lysine-based self-healing hydrogel [136]; (**D**) Polyurethane (with short perfluorobutyl chain) waterproof and breathable membrane [137]; (**E**) Polycaprolactone-based skin-conformable 3D-printed wound dressing [138].

**Table 1 pharmaceutics-13-00341-t001:** Literature search methodology.

	Dermatillomania	Biomaterial Based Therapies	
**Database**	PubMed	PubMed	Google Scholar
**Search date**	April 2020	May 2020	May 2020
**Keywords**	Dermatillomania, excoriation disorder, skin picking disorder, neurotic excoriation, psychogenic excoriation, acne excoriee	Second skin, extra skin, artificial skin, synthetic skin, skin substitute, breathable, polymers, cloth, textile, antibacterial.	Polymer, antimicrobial, biomaterial, on-skin, wearable, aesthetic, water resistant, water proof
**Search method**	dermatillomania[All Fields] OR “excoriation disorder”[All Fields] OR skin-picking[All Fields] OR “neurotic excoriation”[All Fields] OR “psychogenic excoriation”[All Fields] OR “acne excoriee”[All Fields]	1. “second skin”[All Fields] OR “extra skin”[All Fields] OR “artificial skin”[All Fields] OR “Synthetic skin”[All Fields] OR “skin substitute”[all fields] AND (“20 May 2015”[PDAT]: “17 May 2015”[PDAT]) AND (“20 May 2015”[PDat]: “17 May 2015”[PDat]) 2. breathable[All Fields] AND (“polymers”[MeSH Terms] OR “polymers”[All Fields]) AND (“skin”[MeSH Terms] OR “skin”[All Fields]) 3. (((((((“anti bacterial agents”[Pharmacological Action] OR “anti-bacterial agents”[MeSH Terms]) OR (“anti bacterial”[All Fields] AND “agents”[All Fields])) OR “anti bacterial agents”[All Fields]) OR “antibacterial”[All Fields]) OR “antibacterials”[All Fields]) OR “antibacterially”[All Fields]) AND ((((((((“clothing”[MeSH Terms] OR “clothing”[All Fields]) OR “clothes”[All Fields]) OR “clothings”[All Fields]) OR “textiles”[MeSH Terms]) OR “textiles”[All Fields]) OR “cloth”[All Fields]) OR “clothed”[All Fields]) OR “cloths”[All Fields])) AND (“skin”[MeSH Terms] OR “skin”[All Fields]).	Find articles with all of the words polymer antimicrobial biomaterial on-skin wearable aesthetic and with at least one of the words water-resistant waterproof appearing anywhere in the articles
**Results**	439	963	-
**Screening strategy**	All types of articles including case studies, research papers and review articles were referred to.	Title/abstract was read and suitable research articles with products having properties similar to the listed ideal properties were chosen accordingly.	Title/abstract was read and suitable research articles with products having properties similar to the listed ideal properties were chosen accordingly

**Table 2 pharmaceutics-13-00341-t002:** Required properties and functions for an ideal physical barrier product.

Ideal Physical Barrier Product	
Properties	Functions
Biocompatible	Non-toxic and non-allergenic on topical application to skin.
Wearable, waterproof, detergent resistant and easily removable	To prevent skin picking, consciously or subconsciously, at all times and allow already damaged skin to heal. To allow normal functioning of body without interfering with daily life activities.
Skin camouflaging or aesthetically appealing	To prevent attention seeking and improve psychological and social quality of life.
Mimicking mechanical properties of skin	Mechanical strength to resist tearing (product acting as a substitute to skin) caused by skin picking behavior.
Breathable	To allow optimal transepithelial water loss or MVTR, to promote wound healing and prevent skin maceration (causing uneven and easily peelable skin) which can be a trigger for skin picking.
Anti-microbial	To prevent microbial infections in the already damaged skin.
Self-healing material	To allow reuse of product if mechanically damaged by skin picking behavior.
Biodegradable	To prevent environmental pollution.
Cost-effective	To allow affordable, regular usage due to the chronic and recurrent nature of skin picking.

**Table 3 pharmaceutics-13-00341-t003:** Important properties of skin.

Properties of Skin		References
**Skin structure and thickness**		
Stratum corneum	14 μm	[123]
Epidermis	20–150 μm	[123]
Dermis	1–4 mm	[123]
Hypodermis	>1 mm	[123]
**Mechanical properties of human skin**		
Tensile strength	5–30 MPa	[124]
Elastic modulus	0.42 to 0.85 MPa from torsion tests 4.6 to 20 MPa from mechanical equipment 0.05 to 0.15 MPa from suction tests	[125] [124] [126]
Fracture strain	140–180%	[125]
Tear energy (fracture toughness values) by scissors	1700–2600 J/m^2^	[122]
**Other properties**		
Moisture vapor transmission rate (MVTR)	10–75 g/m^2^/h	[127]
Skin roughness	R_z_ (Average roughness of skin surface) = 84.3 ± 12.3 μm R_a_ (difference between the tallest “peak” and the deepest “valley” in the surface) = 6.7 ± 0.6 μm	[128]

**Table 4 pharmaceutics-13-00341-t004:** Biomaterials or products in research which may be adapted for treating skin picking disorder (SPD).

Product and Production Technique	Components	Properties	Intended Use	Suggested Future Studies or Improvements for Use in SPD	Upscale	Ref.
Emulsion: Two step emulsification forming an in situ crosslinked polymer layer. Addition of aqueous phase to silicone phase in a mixer and homogenization. Emulsion 1 applied on skin first and then emulsion 2. Platinum catalyzes crosslinking polymer layer by hydrosilylation	Emulsion 1: Water-in-silicone emulsion with polysiloxane reactive polymer blend (vinyl dimethicone and hydrogen dimethicone) containing 27% (*w/w*) fumed silica in continuous phase Emulsion 2: water-in-silicone emulsion with platinum catalyst (200 ppm *w*/*w*) and nylon 10–12 in continuous phase	* In situ polymer crosslinked in 2 minutes * Mechanical properties: elastic modulus = 0.48 MPa, fracture strain = 826%, adhesive strength = 78 N/mm, elastic recoil with minimal strain-energy loss. * Thickness of film = approx. 40 μm. * Biodegradable and biocompatible * Wearable for up to 16 hours, easy removal without damaging skin and breathable. * Polymer film intact even with daily activities like swimming and running * Water-resistant and detergent resistant, rub and wash resistant. * Aesthetically appealing. Gives appearance of natural skin.	Restores compromised skin barrier function; Can be used for pharmaceutical delivery and wound dressings. Used in successfully treating AD patients as an adjuvant treatment [132]	-Incorporating nanoparticles in the film can be a means to combine other actives such as antimicrobials, vitamins, wound healing factors and nutrients. -Improving the current once daily application mode	Feasible	[125]
Emulsion: One step emulsion system forming in vivo crosslinkable polysiloxane coating. Preparation of catalyst capsules dispersion by solvent evaporation method. Addition of aqueous phase to silicone phase in a Mixer and homogenization to obtain V and H emulsions. K,V,H parts are blended in 0.25/9.1/0.9 ratio. These three parts are isolated from each other by a continuous water phase.	(K) Karstedt (Pt) catalyst capsules dispersion (K) (V) Vinyl dimethicone emulsion = 30% *w/w* (V) (H) Hydrogen dimethicone emulsion = 30% *w/w* (H)	* In situ cross-linked polymer formed with tack free time of 10–30 min * Tensile strength = 0.55 MPa, elongation at break = 250%, elastic modulus = 0.47MPa * Thickness of dried film = 50 μm * Biocompatible and safe * Skin adherent and wearable * Easy single step application * Comparable to WVTR of human skin * Gives appearance of natural skin. * Water proof and high adhesion strength to human skin and also can be peeled off without irritating or harming the underlying skin.	Suggested as base materials for dermatological drug carrier, wearable electronic skin and wound dressing.	Incorporation of nanoparticles in the film to introduce antimicrobial and wound healing activities. * More studies on daily wear and wear time.	Feasible	[127]
Spray based Suspension Addition of LL37-SH to citrate@AgNPs and incubation followed by crosslinking of type1 collagen with addition of glutaraldehyde. Addition of excess glycine to quench glutaraldehyde.Final formulation has a total silver concentration of 100 um	Type 1 medical grade collagen, LL37-SH (antimicrobial peptide) Citrate capped Silver nanoparticles	* Non toxic * Antimicrobial properties (*P.aeruginosa*) [133] *Staphylococcus aureus* [134]. * Silver NP have a wide spectrum antimicrobial property. * Sprayable on wounds * Remains in place when sprayed into skin wound * Minimal organ infiltration upon spraying on wound.	Spray-on topical application for prophylactics and infection control in infected wounds	In addition, this technology maybe developed for spraying on clothes or products in contact with skin, to achieve antimicrobial properties and prevent infections.	Feasible	[133]
Hydrogel: Polyelectrolyte and self-healable. One-step random copolymerization of AA and DMAPS monomers	Acrylic acid (AA) and 3-dimethyl(methacryloyloxyethyl) ammonium propane sulfonate (DMAPS)	* Viscoelastic behavior with solid like elasticity and liquid-like plasticity * Imitates mechanical properties of natural skin Wide spectrum time-dependent mechanical properties with Compressive modulus of 27.6 KPa * Flexible reconfiguration ability: Can be reconfigured to fabricate a thin layer of transparent hydrogel skin. Can be adapted to irregular surfaces and was shown to be compliant with prosthetic finger locomotion. * Robust elasticity * Extremely stretchable: can be stretched more than 10000% the original length without fracture elongation of >100 without fracture * Fast autonomous self-healable within 2 hours. * Recyclable: It can recover >90% G’ in 10 dehydration-hydration cycles.	Used on prosthetic finger to sense train and temperature stimuli through capacitive and resistive sensors respectively. To be used to construct deformable sensory systems in the next generation of soft intelligent robots and smart wearable devices for IoT applications.	This technology may be improved to form self-healing patches or apparels that can be stuck at regularly skin picking areas which may be helpful. Long term wearability and biocompatibility on skin to be assessed.	Feasible	[135]
Hydrogel:Enzyme-induced dual-network EPL based hydrogels Self-healing (EDH). Radical polymerization of NVP and NMA under EPL to form single network EPL-G-POLY(NVP-*co*-NMA) hydrogels. Followed by Plasma amine oxidase (PAO) catalyzing in situ Schiff base reaction to form double network hydrogel	1-vinyl-2-pyrrolidinone (NVP) N-methylol acrylamide (NMA) Epsilon-poly-l-lysine (EPL)	* Biocompatible * Self-healing synthetic material. High autonomous self-healing efficiency of 95% without any external stimuli * Broad spectrum antimicrobial activity against both Gram-negative and Gram-positive bacteria. * Enhances wound healing with minimal inflammatory response. Wound closure rate of 97% * Robust mechanical strength ~0.11 MPa * EPL exhibits potential adhesive property	Suggested use and great potential in myriad biomedical fields, such as wound repair, artificial skin and tissue engineering	May be developed into patches or films for application over picked skin for wound healing and protect that area from being picked by individuals consciously or unconsciously.	Feasible	[136]
Fibrous membrane Electrospinning (of PU/C4FPU/AgNO_3_ in N, N-dimethylacetamide)	Polyurethane elastomer (C4FPU) possessing double terminal short perfluoro butyl (−C_4_F_9_) chain Polyurethane (PU) Silver nitrate (AgNO_3_)	* Eco-friendly * Water proof (water resistant property of 102.8 kPa) * Breathable (WVTR of 12.9 kg.m^−2^·d^−1^) * High mechanical property of 9.8 MPa * Anti-bacterial activity (against *S. aureus and E. coli*)	Suggested for developing protective garments/textile	More studies on alteration of properties with respect to wash-reuse cycles to develop into aesthetic apparels. Incorporation of nanoparticles into nanofibers for other desired functions.	Feasible	[137]
3D printed wound dressing 3D scanning of physical object or body part and 3D printing of wound dressings using prepared silver-loaded PCL filament, copper-loaded PCL filament zinc-loaded PCL filament	Polycaprolactone Silver nitrate Copper sulphate (II) pentahydrate Zinc oxide	* Biocompatible and biodegradable * Flexible due to elastomeric properties of PCL * Personalized treatment: Personalized wound dressings anatomically adaptable * Bactericidal properties of Silver loaded PCL dressing and copper loaded PCL dressing * Dressings can be tailored to shape, size and with antimicrobial agents.	Customizable wound dressing	Evaluation of safety and wearable time for this type of wound dressing. Can be used for developing patches or other apparels to promote wound healing and prevent bacterial infections.	Feasible	[138]

**Table 5 pharmaceutics-13-00341-t005:** Commonly used polymers for developing skin-mimicking materials.

Type	Examples	Reference
Natural polymers	Collagen, hyaluronic acid, chitosan, gelatin, elastin, pullulan, alginate, dextran, cellulose, agar, agarose, carrageenan, pectin, keratin, fibrin, silk fibroin, egg shell membrane, Heparin	[123,140,141,142]
Synthetic polymers	Polyurethane, poly (l-lactic acid)(PLLA), poly(glycolide-*co*-l-lactide) (PLGA), poly(ethylene glycol) (PEG), polycaprolactone (PCL), poly(N,N-diethylacrylamide), poly(N-vinyl-2-pyrrolidone), polyvinyl alcohol (PVA), polyacrylic acid (PAA), silicones (polydimethylsiloxanes)	

## Data Availability

No new data were created or analyzed in this study. Data sharing not applicable.

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
