# Peer review of "Dermatillomania: Strategies for Developing Protective Biomaterials/Cloth"

_pharmaceutics, 2021, doi:10.3390/pharmaceutics13030341_

Round 1

Reviewer 1 Report

Review Priusha et al. “Pharmaceutics”

The paper titled “Dermatillomania: Strategies for developing Protective Bio-materials/Cloth” presents Skin-Picking Disorder (SPD) and several treatment strategies currently used or under investigation. The paper focuses mostly on physical barriers and skin replacement materials, including some incorporating nanoparticles.

The review is interesting and relatively well written. I recommend the paper be checked again by a native English speaker as there are still many mistakes and awkward phrasing (I have listed a few examples below).

As the main findings are summarized in Table S2, why is this table not in the main text?

Furthermore, I think having more figures to show the chemical structures of the materials under discussion would help (for instance l420-421: “a novel polyurethane elastomer”; l374: “a AgNP@LL37 collagen film”; etc)

Minor comments and questions are listed below:

L20-21: this sentence is difficult to understand and should be rephrased.

L49: “damage they” > “damage they”

L90 and 171: “cope up with” > “cope with”

L97: “the vice versa” > maybe “the opposite” sounds better

L123: “few of them”: do you mean “a few of them”?

L172: “to be reduce” > “to reduce”

L183: “such as SSRIs” > “such as Selective Serotonin Reuptake Inhibitors (SSRI)”

L209: “has been” > “have been”

L323, 325 and elsewhere: “maybe” > “may be”

L576: is the dmax unit micrometers?

Supplementary material, P20: “Mimitates” should probably be “Imitates”

Reviewer 2 Report

The topic is new on Dermatillomania and futuristic and no similar thematic review articles have been published till date. Does correlate and give significant information on Dermatillomania related prevalence, etiology, consequences, diagnostic criteria, treatment strategies especially on limited therapeutic strategies. However, extensive English language revisions are required. The review certainly add scrutiny in the respective field of research. Further, few minor revisions are required in order for the review to be accepted for publication.

1.      Abstract does not highlight why studies on Dermatillomania related therapies are important. Even does not discuss the future research or outlook of this type of study.

2.      A new section -Dermatillomania as a psychodermatologic disorder should be created and briefed with case studies.

3.      A table is recommended for grouping the methodology or experimental design- separately for Dermatillomania and biomaterial based therapies.

4.      The discussions are not example based does not connect properly with just directional explanation. Authors are advised to add certain relevant examples so that readers get a detailed information.

5.      A table listing the novel drug delivery systems and biomedical engineering products such as scaffolds, decellular tissues being prepared to overcome Dermatillomania are reported in literature in recent times and these need to be added and is strongly recommended for the review.

6.      The review lacks upbringing the importance of irrational use or medication under physician direction concept to minimize Dermatillomania and their importance in highlighting to build and safe use of existing therapies. New references suggested below should be included in the discussion.

7.      A new section or table on listing all the Pharmaceutical and Biomedical formulations and their possible benefits along with route of administration should be inserted.

8.      A detailed explanation on how this correlation is important for different stakeholders such as healthcare professionals, patients, scientist and pharmaceutical or biomedical industries needs to be commented in the conclusion section.

9.      The review lacks extraction of valuable figures or data from the references cited. Authors are strongly suggested for the needful.

10.  Also below are the few latest papers /case studies on Dermatillomania and its novel delivery systems or biomaterial or biomedical developed and evaluated, authors are strongly advised to cite and mention in the respective discussion sections. Authors are strongly suggested for these recommendations to be considered for further evaluation.

In Atlas of Dermatoses in Pigmented Skin (pp. 627-636)

 International Journal of Biological Macromolecules. 2020 Oct 15.

Journal of Child and Adolescent Psychopharmacology 30, no. 10 (2020): 580-589.

European Polymer Journal (2020): 109919.

Aktuel. Dermatol. 43, no. 11 (2017): 477-492.

Journal of Engineered Fibers and Fabrics 10, no. 4 (2015): 155892501501000411.

Quantitative imaging in medicine and surgery 7, no. 1 (2017): 166.

DERMATILLOMANIA: A CASE OF EXCORIATION DISORDER." (2019).
